# Work Hardening of Heat-Treated AlSi10Mg Alloy Manufactured by Single and Double Laser Selective Laser Melting: Effects of Layer Thickness and Hatch Spacing

**DOI:** 10.3390/ma14174901

**Published:** 2021-08-28

**Authors:** Emanuele Ghio, Emanuela Cerri

**Affiliations:** Department of Engineering and Architecture, University of Parma, Via G. Usberti, 181/A, 43124 Parma, Italy; emanuele.ghio@unipr.it

**Keywords:** work hardening, layer thickness, AlSi10Mg, Selective Laser Melting, mechanical properties

## Abstract

The present study analyzed the microstructure and the mechanical properties of AlSi10Mg SLMed bars (10 × 10 × 300 mm) and billets (10 × 100 × 300 mm) before and after the direct aging at 200 °C for 4 h and the T6 heat treatment. The discussed results are compared to those obtained by the AlSi10Mg samples manufactured with the same geometry but using different process parameters (layer thickness higher than 40 μm and a hatch spacing lower than 100 μm) and also through the Quality Index (QI). These work conditions allow the obtaining of a microstructural variation and different tensile properties in as-built top samples. In both batches, the cycle time was 45 h and together with the preheated build platform at 150 °C, induced an increase of UTS (Ultimate Tensile Strength) and yield strength on the bottom rather than the top samples due to the aging phenomena. Upon completion of the direct aging heat treatment, the effects induced by the platform were cancelled, keeping a full cellular microstructure that characterized the as-built SLMed (Selective Laser Melted) samples. Moreover, the Considère criterion and the work hardening analysis showed that the failure occurs after the necking formation in some of the T6 heat-treated samples. In this last case, the Si eutectic network globularized into Si particles, causing a decrease of UTS (from around 400 MPa to 290 MPa) in favour of an increase of ductility up to 15% and reaching a QI in the range 400 ÷ 450 MPa. These values place these samples between the high-quality aluminium cast alloy and T6 heat-treated ones.

## 1. Introduction

The Selective Laser Melting (SLM) process allows for the obtaining of a 3D physical object, with complex geometry, built by a layer-by-layer method. Due to this machine set-up, the process takes place in the powder bed fusion category within the additive manufacturing field. During the print process, a laser beam scans a single layer (thickness in the range between 20 and 100 μm) according to the CAD (computer-aided design) project [1]. In this context, the great customization capabilities that characterize this process, and the good performance obtained by fabricated parts compared to the conventional manufacturing process, are controlled by the optimization of different process parameters [2,3,4]. In other words, through the perfect combination between the scan strategy and the energy density (φ, [J/(mm^3^)]), which can be defined as follows:φ = P/vht(1)
where P is the laser power [W], v is the scan speed [mm/s], h is the hatch distance [mm] and the t is the layer thickness [mm]; moreover, the denominator of the fraction is defined as build rate (BR = vht, [(mm^3^)/s]) [5,6,7]. On the other hand, using a higher hatch distance or a combination of low laser power with high scan speed, or powder containing gas, the density and the consequent mechanical properties of the manufactured part are compromised [3,4,8,9].

Considering instead the layer thickness (t) and focusing on the influence on mechanical properties induced by its variation, many studies of this nature cannot be found in the literature. The research of Yastroitsev et al. [10] places the layer thickness as the second parameter in terms of importance after laser power. In fact, the process parameters of the SLM process should be recalibrated after the layer thickness variation. First, the deposition process, the packing density and the homogeneity of a bed powder layer can be compromised [11]. Second, with an increase of t the increase in depth of penetration is necessary to ensure the adhesion between the new scanned layer and the previously solidified layer. Sufiiarov et al. [12] and Nguyen et al. [13] studied the Inconel 718 alloy and a decrease in tensile strength combined with increasing layer thickness from 20 to 50 μm resulted. The other outcome was the variation in terms of BR (Build Rate); in fact, the volume of powder bed scanned increases with an increase of t decreasing the process time.

In this context, a reduction in the processing time can also be obtained by a multi-laser machine setup, and in this scenario a few types of research can be found in the literature [14]. As matter of fact, Liu et al. [15] showed that the mechanical properties of Ti6Al4V alloy are uniform between the samples produced by a single and a multi-laser. Moreover, the use of a multi-laser influences the molten pool, the HAZ (heat-affected zone) and the thermal gradient, as reported by Masoomi et al. [16] and B. Liu et al. [17] for the Ti6Al4V and AlSi10Mg alloys, respectively. As concerns the aluminium alloy, C. Zhang et al. [14] demonstrated slight microstructure differences and the consequent slight decrease of UTS between 481 ± 2 MPa for single laser samples and 471 ± 7 MPa for multi-laser samples. Finally, focusing on the AlSi10Mg, the alloy selected in the present paper, there are many studies concerning only the influence on the mechanical properties and the density caused by the variation of scan speed, hatch distance, laser power and scan strategy [3,4,9,18,19].

The AlSi10Mg alloy is widely used in the SLM process and its near-eutectic composition of Al and Si make the process possible by laser beam [19]. Considering the optimization of previous process parameters and the high cooling rate (10^6^ ÷ 10^7^ K/s) that characterizes the SLM process, both a very fine microstructure and a fully dense part can be obtained. Moreover, the full-cellular structure of the Si eutectic network in the α-Al matrix that is formed, in addition to the Orowan looping behaviour, make the mechanical properties of the AlSi10Mg SLMed higher than the same casting alloy. As matter of fact, the SLMed samples show YS between 250 and 300 MPa, UTS ≤ 400 MPa and strain at failure ≤7% rather than YS between 90 and 110 MPa and UTS between 190 and 210 MPa of casting alloy [4,8,19,20,21,22,23,24,25]. Additionally, the use of a preheated build platform usually in the range between 150 and 200 °C can increase the mechanical properties of as-built samples due to the aging phenomena [26]. This range of temperature causes the precipitation of Mg2Si along the boundary of α-Al due to SSS (supersaturated solid solution) that is generated during the solidification of the molten pool [23,26,27,28]. Casati et al. [26] analyzed the effects induced by the preheated build platform on AlSi10Mg samples SLMed. The authors show that the DSC scan profiles highlighting that the precipitation phenomena occurred during the “hot-stage” manufacturing process.

Therefore, starting from an excellent manufactured part, it is possible to perform different post-processing heat treatments for an improvement of compromise between tensile strength and elongation at break. This Al-Si-Mg alloy is mostly used in automotive and aerospace industries, and the optimization of post-process heat treatments can implement the areas of use. For example, the SLM process induces high residual stress due to the thermal effects in as-built samples. Therefore, stress relief or new heat treatment are needed to reduce this possible cause of failure during use. In this context, it is possible to emphasize that the improvement of ductility is the main aim in the AM field. Substantially, the scope is an increase of the elongation at break without decreasing the ultimate tensile strength (UTS) and yield strength (YS). In this context, the T6 heat treatment (solution heat treatment “SHT” + water quenching “WQ” + artificial aging “AG”) causes a good improvement in ductility (≈20%) and a drastic decrease of the UTS [29,30]. The SHT allows obtaining a dissolution of soluble phases or traces of other elements, but the high temperature induces a globularization of Si particles, which is the main cause of loss in terms of UTS and YS, as previously reported [30]. Subsequently, only after the water quenching the microstructure is stabilized at room temperature. In fact, considering the AlSi10Mg phase diagram, the solubilization of the element within the α-Al matrix would decrease with a slow decrease in temperature. Finally, a recovery in terms of mechanical properties is caused by the precipitation (GP zones → β″ → β′ → β) of the β-Mg2Si phases during artificial aging [31,32]. At the same time, the good compromise between the tensile strengths and the strain at break can be obtained by direct aging (DA), i.e., by heat treatment performed directly in as-built samples without the SHT. The DA is normally carried out at a temperature lower than 200 °C and can maintain the full-cellular structure that characterized the as-built samples SLMed [23,26,33,34]. Cerri et al. [23], analyzing the AlSi10Mg samples, showed a gradual destruction of the Si eutectic network after the DA was at 225 °C for 6 h. Nevertheless, the obtained mechanical properties were satisfactory (UTS of 350 ± 10 MPa, YS of 200 ± 8 MPa and ε 14 ± 1%).

This paper aims to highlight the effects induced by the variation of the layer thickness (t) and of the hatch spacing (h) in the AlSi10Mg alloy manufactured by single and multi-laser selective laser melting. The work was conducted through optical microscopy analysis. Vickers microhardness and tensile tests following the Considère’s criterion on samples before and after the DA (200 °C/4 h) and the T6 HT were undertaken. Finally, the quality indexes have been used to evaluate the best compromise between the UTS and the elongation at break values.

## 2. Materials and Methods

AlSi10Mg gas-atomized powder with normal distribution between 20 μm and 60 μm was used and its chemical composition is shown in Table 1.

Figure 1 shows the bars (300 × 10 × 10 mm) and the billets (300 × 100 × 10 mm) manufactured by SLM280 equipped with 400W IPG fibre laser. The yellow and red parts of the preheated build platform at 150 °C are the single laser (SL) and double laser (DL) zones. For clarity, all samples and bars were be divided into “bottom” (height < 150 mm) and “top” (height > 150 mm) regions. Moreover, both the tensile samples and the bars were designated with “SL-90 and DL-90” rather than “SL-50 and DL-50” of those analyzed [23] and considered in the present study. In this context, 90 and 50 referred to the different layer thicknesses adopted during the SLM process, while the other process parameters are shown in Table 2.

According to the ASTM E804 standard, 26 tensile samples (from 1 at the bottom to 26 at the top region) have been machined into machine tools for each billet. Their symmetry axis was either perpendicular to the build direction (Z-axis) or parallel to the build platform, as shown in Figure 1a.

The AlSi10Mg samples were analyzed in as-built condition and after two different heat treatments. The first was the direct aging at 200 °C/4 h; the second was the T6 heat treatment (HT) (505 °C/4 h + 175 °C/4 h) as shown in Table 2.

The tensile tests were performed at room temperature using a Z100 Zwick/Roell servo-hydraulic device and at a strain rate of 0.008 s^–1^ in accordance with ASTM E804.

The Vickers microhardness was measured along both the XZ plane and the XY plane in all heat treatment conditions using a 500 gf load and a dwell time of 15 s. The profiles along the XZ plane were performed both on the external frame and the centre of the material. In both cases, they were developed along different lines parallel to the edges of the bars, and each of these was formed by 60 indentations spaced 5 mm apart. The Leica VMHT microhardness tester (Leica, Wetzlar, Germany), which is equipped with a standard pyramidal indenter, was used for all measurements discussed in the present study.

The microstructure was analyzed by the DMi8 optical microscope (OM) (Leica, Wetzlar, Germany), and by different Vickers microhardness profiles with 15 gf of load in as-built condition.

The density of all samples was obtained through the 2D analysis of pores performed on the samples’ surface, which was mechanically ground by SiC paper and then polished with colloidal suspension. The analysis was carried out with the Leica OM equipped with LAS-X 2D software (LAS AF V4.0, Leica Microsystems, Mannheim, Germany). According to ISO 13322-1:2014, six micrographs (at a magnification of 100×) were acquired and, subsequently, analyzed to determine the total pores area. Therefore, using the image analysis method, the density $\left(\rho \right)$ was calculated as follows:(2)$$\rho =1-\frac{\sum_{i}{\left(A_{p}\right)}_{i}}{\sum_{j=1}^{6}{\left(\tilde{A}\right)}_{j}}=1-\frac{\sum_{i}{\left(A_{p}\right)}_{i}}{24936.41{\;\mu m}^{2}}$$
where $\sum_{i}{\left(A_{p}\right)}_{i}/\sum_{j=1}^{6}{\left(\tilde{A}\right)}_{j}$ is the ratio between the total area of pores and the total area analyzed was 24,936.41 μm^2^.

## 3. Results

### 3.1. Microstructure

Figure 2 shows the characteristic microstructure of SL-90 and SL-50 AlSi10Mg SLMed samples considering the XZ (first row Figure 2a,c,e) and the XY (second row Figure 2b,d,f) planes, respectively. Considering Figure 2a, the typical fish-scale structure formed by the molten pools’ cross-sections can be seen. This particular structure is located in the build direction due to the laser handling during the layer (n) scanning, which also remelts a part of the previously solidified layer (n − 1). On the other hand, the depth of the cross-section and the overlap area are controlled by density energy (φ) and by hatch spacing, respectively [35]. Only after the analysis of the XY plane can the characteristic ellipsoidal shape of the molten pool be observed (Figure 2b).

Focusing on Figure 2c,d, it is possible to highlight the complex thermal history which characterized the samples manufactured through the SLM process. As matter of fact, the high cooling rate (10^6^ ÷ 10^7^ K/s), various heat fluxes, and the different thermal gradients induce a variation in grain dimensions and morphology [17,26,36]. In this context, the molten pools’ cross-sections are often formed by columnar grains that are arranged along the thermal gradient [37]; however, focusing on Figure 2a, it is possible to observe the presence of both the columnar and equiaxed grains into different molten pools. Figure 2c shows a high magnification micrograph of the molten pool cross-section (XZ plane) where the yellow dotted line highlights its boundary. The only presence of equiaxed grains that dominate the entire microstructure can be observed. Pham et al. [37] and Guan et al. [38] showed the same microstructural configuration through the EBDS analysis carried out on AlSi10Mg SLMed samples. On the other hand, the SL-50 samples show the molten pool cross-sections (Figure 2e) that are only dominated by columnar grains arranged along the thermal gradient, as studied and reported in [23]. Lingda et al. [39] show how the solidification process is described by a planar growth followed by a competitive grain growth between the equiaxed and columnar grains within the molten pool due to the differences in undercooling, which generates an instability at the interface between liquid and solid. With an increase of the undercooling, the molten pool cross-section is totally formed by equiaxed grains, which show a decrease in size from the fusion line to the center of the molten pool. A. Hadadzadeh et al. [40] demonstrated that the CET (columnar-to-equiaxed transition) is strictly related to the thermal gradient (G) and the solidification rate (R) ratio; in fact, if the G/R decreases the CET is promoted. Moreover, as reported by the same authors, the crystallographic orientation is the other important parameter that influences the CET due to the correlation between the undercooling and the angle between the growth rate and <hkl> direction of the growing dendrite. Finally, it can be emphasized that the analyzed 90-samples are characterized by an inhomogeneous solidification process along the different planes considered, as shown in Figure 2d, where a magnification of Figure 2b highlights the area between the molten pool and the laser scan tracks. In this case, the columnar grains characterize the microstructure along the XY plane, unlike the SL-50 sample (Figure 2f) where the microstructure is only formed by the equiaxed grains as analyzed in [23], and as reported by [40]. Normally, the XY plane shows the equiaxed grains within the laser scan tracks or the molten pool where their dimensions increase from the center to the edges due to the heat input produced during the SLM process [41,42,43]. In this context, the process parameters have a considerable influence on the heat input; in fact, as reported by Paul et al. [44], the number of columnar grains increase with increasing the layer thickness or the hatch spacing; while the quantity of equiaxed grains increases and their dimensions decrease with decreasing the hatch spacing. As demonstrated by the same authors, the decrease of hatch spacing (≈−53%) induces a greater influence on microstructure than an increase of the layer thickness (≈+50%).

Figure 3 shows the microhardness profile performed along with the different directions and considering the different kinds of microstructure previously discussed. Figure 3a–d shows the XZ plane and XY plane of SL-90, respectively; while Figure 3e,f show the same planes referred to SL-50 samples. Generally, the microhardness values decrease in correspondence to the molten pool (about 15 and 22 HV15) and scan tracks boundaries (about 36 HV15) focusing on Figure 3a,b,d, respectively, due to the effects induced by HAZ, i.e., the destruction of the Si network and the presence of coarse zone [23,43,45]. As reported by Chen et al. [46], this microstructural configuration is less effective at hindering the dislocation movements; in fact, it shows a greater plastic deformation than the other zones formed by a full-cellular structure. On the other hand, the microhardness profile shown in Figure 3c does not show a great variation between the center and the molten pool boundary. Considering the average HV15 trend, this is lower than those related to the equiaxed grains (Figure 3a,b,d) because the columnar grains are characterized by a lower deformation resistance under load than the equiaxed grains [47,48,49]. As matter of fact, the molten pool cross-section characterized by equiaxed grains (Figure 3a) shows 133 ± 10 HV_15_ rather than 107 ± 4 HV_15_ of the cross-section characterized by columnar grains (Figure 3c). These values are the average of the microhardness obtained in each profile discussed. Finally, comparing the microhardness profile shown in Figure 3a–d to that shown in Figure 3e,f, the SL-90 are characterized by higher HV15 values than the SL-50 due to the different microstructures, which is caused by the increase of layer thickness and the decrease of hatch spacing that have influenced the cooling rates of 1.62 × 10^7^ K/s and 9.57 × 10^6^ K/s, respectively. These values are evaluated after the analysis of the secondary dendrite arms spacing (SDAS) and through the following equation:(3)$$\mathrm{SDAS}=k\dot{T}{}^{-n}$$
where k and n are the pre-exponential (k = 43.2) and exponential (n = 0.324) constant, and $\dot{T}$ is the cooling rate expressed in (K/s), as reported by Matyjia et al. [50]. All of these confirmed by Liu et al. [51] justified the increase of hardness between 50 and 90-samples by the fine grains formed due to the increase of overlap regions. In other words, this was due to the decrease of the hatch spacing. Hyer et al. [52] analyzed the characteristics induced by the variation of different process parameters in AlSi10Mg SLMed samples, and through this showed a decrease in cell size with the increase of scan speed (from 800 to 1800 mm/s) and of the layer thickness (from 30 to 90 μm).

Figure 4 shows the microstructure obtained after the DA at 200 °C/4 h and T6 for AlSi10Mg DL-90 samples. In this context, it is necessary to highlight the perfect comparison between the SL-90 and DL-90 in terms of microstructure. As reported by Cerri et al. [23], there is no microstructure variation between the microstructure of SL-50 and DL-50 both in the as-built case and after the DA at 200 °C/4 h. In this context, it is possible to observe the same result comparing the microstructure shown in Figure 4a,b to those reported in Figure 2. On the other hand, the T6 heat-treated samples show a greater microstructure variation. The as-built AlSi10Mg samples are characterized by a solid supersaturated solution of Si in the α-Al matrix due to the high cooling rate and the Si eutectic that forms a fibrous structure which will be destroyed after the SHT. Only after the SHT and artificial aging process is the Si rejected from the α-Al matrix and the Si particles precipitate along the Al-Si boundaries.

Increasing the SHT temperature causes the Si particles to sgrow, and their number decrease as a consequence [36]. Simultaneously, the Mg2Si-phase precipitates and the elongated and brittle β-Al5FeSi phase grows in α-Al matrix as reported by [42,53].

In terms of Si eutectic (Figure 4d), the microstructure of the SL-50 samples after the T6 heat treatment is shown in Figure 4d, and it is possible to observe the same microstructural conditions of the DL-90. Finally, the ellipsoidal shape of the molten pool is still visible in the XY plane (Figure 4d) due to the particle accumulation effect along the edges of scan tracks rather than at their centre, as reported by Y. Li et al. [36].

### 3.2. Mechanical Properties

Figure 5a,b shows the Vickers microhardness profiles performed along the Z-axis on the external frame (red profiles) and the CM (orange profiles) of the SL-90 and DL-90 bars in as-built condition, respectively. Instead, the two blue curves represent the microhardness profiles of as-built SL-50 and DL-50 measured along the build direction in the CM. Considering the orange and blue microhardness profiles in Figure 5a, both the SL-90 and the SL-50 show the same effects of aging phenomena induced by the preheated build in the bottom region rather than in the top region, which is strengthened only by solid solution, as reported by [23,27]. In fact, the HV500 values decrease from 124 ± 3 HV500 to 109 ± 3 HV500 and from 130 ± 2 HV500 to 114 ± 4 HV500 for SL-50 and SL-90, respectively. The average difference of the microhardness values (~6 HV500) between the SL-90 and SL-50 can be attributed to the difference in grain size dimension as previously discussed in Section 3.1. On the other hand, the variation between the CM (orange profile) and the external frame (red profile) is caused by the difference in terms of density as reported in Table 3. Comparing the bottom and top values, it is possible to observe that the variation of density remains constant (1.88% and 2.04%, respectively).

Figure 5b shows the same trend previously discussed for CM SL-90; in fact, the microhardness values decrease from 130 ± 2 HV500 to 117 ± 3 HV500. Focusing on the microhardness profile of the DL-90 external frame, the effects induced by the aging phenomena are reduced by the high density of the pores, as illustrated by the density values shown in Table 3.

In fact, the difference in density in the bottom region between the CM and the external frame is around 12.20%. In the same context, as reported in [23], the as-built DL-90 shows lower values than the as-built SL-90. Moreover, if the density of SL samples decreases along the build direction, the inverse behaviour characterizes the DL samples. Andani et al. [54] attribute the density decrease of DL samples to the unmelted spatters phenomena, which is less present in SL samples. Moreover, it can be highlighted that porosity behaviour is not controllable due to the high number of parameters that can influence the density [55]. Chen et al. [56] show the influence of the energy density (φ) on the large pores rather than the small; also, the wrong optimization of all parameters contained in (φ) can induce a high number of defects. The same undesirable results are obtained due to the presence of gas within the gas atomized powder and due to the hydrogen solubility into the melt pool. In this context, the pores nucleation takes place only when the maximum hydrogen solubility is reached in these points. Weingarten et al. [9] demonstrated an increase in pores if they nucleate at the solidification front rather than at the melting front due to the combination between the solidification time and the terminal velocity of pores. The former depends on the ratio between the molten pool length and the scan speed, the later on gravity acceleration, kinematic viscosity and pore radius. Finally, the density can be affected by the scan strategy despite the optimization of (φ) and by Marangoni flow, as reported by [25,57]. Finally, comparing the variation between the maximum 99.87 ± 0.01% and the minimum 98.50 ± 0.06% density values of CM as-built samples with those reported in the literature, the process parameters have been effectively optimized. In this context, it is necessary to observe that only the CM will be subject to the uniaxial tensile load due to the removal of the external frame by machining tools (Section 2). Therefore, the high quantity of pores present in the external frame will not contribute to the causes of the failure of the sample.

After the T6 heat treatment, the densities decrease (see Table 3) due to the pore variation, which can be caused by the variation in grain structure [25,58,59]. Other authors have shown that with the increase of SHT temperature the gas pressure increases and, if the temperature is able to reduce the yield strength of the material around the pore, it can deform it [9,25]. It is certainly possible to say that the pores may spheroidize to minimize the free surface energy as expressed by the Young-Laplace equation [22]. On the other hand, some other authors do not highlight the variation in density after the T6 HT [60,61]. Ultimately, the DA at 200 °C/4 h does not induce any variation in density, as can be seen in Table 3.

As concerns the XY plane, the Vickers microhardness shows a slight difference between SL-90 and DL-90 both in the bottom (132 ± 3 HV500 and 131 ± 3 HV500) and in the top regions (115 ± 5 HV500 and 110 ± 2 HV500). Comparing these HV500 values with those obtained along the XZ plane, it can emphasize the microstructure homogeneity along the different planes as supported by the optical microscope analysis (Figure 2). The same results are shown by Maamoun et al. [42] which analyzed the effects induced by the process parameters on the AlSi10Mg SLMed samples.

Figure 6 presents the tensile properties of AlSi10Mg samples in as-built condition (Figure 6a,b) after the DA at 200 °C/4 h (Figure 6c) and the T6 HT (Figure 6d) were analyzed along the build direction.

Figure 6a shows the mechanical properties of the as-built SL-90 and DL-90, which do not present marked differences in terms of strengths and elongation at break, as also reported by [23]. For these reasons, in the following discussion of heat treatment, only SL-90 and the SL-50 will be considered. The average UTS values decrease from 438 ± 13 MPa and 418 ± 9 MPa to 411 ± 8 MPa and 392 ± 11 MPa for SL-90 and DL-90, respectively. Simultaneously, the yield strengths decrease from 284 ± 10 MPa and 281 ± 9 MPa to 237 ± 6 MPa to 232 ± 4 MPa for the same analyzed samples. The ductility remains constant at around 6.9 ± 1.0%. Figure 6b shows the same decreasing trends of mechanical properties of as-built SL-50 and DL-50 samples that were studied in [23]; but, comparing these to the Figure 6a, it is possible to observe the greater decrease of all tensile strengths along the build direction (Z-axis), as also shown in Table 4, which illustrates the absolute differences.

If the bottom samples are characterized by the same tensile strength values, the top SL-90 and DL-90 show higher UTS and yield strength values than the SL-50 and DL-50. As also reported by [23], the microstructure between the top and bottom regions does not show a variation except for the precipitation phenomena induced by the preheated build platform. This behaviour is also confirmed by R. Casati et al. [26]. In the same context, the temperature used for heating the substrate (150 °C) may have induced a reduction of the residual stress as reported by L. Wang et al. [62]. Therefore, the top region can be affected by higher values of residual stress than the bottom region due to the different residence time within the build chamber during the SLM process, as shown by the top-X axis (Time [h]). Moreover, the variation of tensile strengths between the top 90-samples (Figure 6a) and the 50-samples (Figure 6b) can be justified through the different process parameters used, which induce different residual stress, i.e., differences in terms of dislocation density due to the high cooling rates and the complex thermal cycling, as reported by D. Tomus et al. [63]. In the same context, as reported by M. Yakout et al. [64] the increase of residual stress with decreasing of hatch distance and increasing energy density is shown, i.e., the same conditions that differentiate the manufacturing process between the 90-samples and the 50-samples (Section 2).

Figure 6c shows the tensile properties of SL-90 and SL-50 after the DA at 200 °C/4 h. The DA induces a greater homogenization between the bottom and top regions for the SL-90 than the SL-50 considering the absolute variation shown in Table 3. In this context, the average UTS values decrease from 373 ± 3 MPa and 369 ± 8 MPa to 363 ± 7 MPa and 348 ± 2 MPa for SL-90 and SL-50, respectively. Simultaneously, the yield strengths decrease from 230 ± 3 MPa and 229 ± 6 MPa to 219 ± 6 MPa to 210 ± 2 MPa for the same analyzed samples. The ductility remains constant at around 8.3 ± 0.9%, which is comparable to that shown by the as-built samples. In addition, the top regions of SL-90 were characterized by higher strength values than the SL-50. This behaviour can justify the argument about the residual stress because the DA at 200 °C/4 h can tend to reduce it like a preheated build platform; moreover, it does not influence the microstructure (Figure 2 and Figure 4), as also reported in [23,33].

On the other hand, the T6 heat treatment induces a significant decrease of the UTS values (≈280 MPa), despite the YS values remaining constant to those shown after the DA heat treatment. Moreover, the absolute differences of the tensile properties between bottom and top samples are substantially knocked down for the SL-90; while for SL-50 the UTS and YS values show a slight increase from the bottom to the top regions. Finally, the average elongations at break increase up to 9.5 ± 1.2% and 10.6 ± 2.4% for SL-50 and SL-90, respectively. All of this is attributable to the microstructural variation shown in Figure 4.

As widely reported in the literature, the strengthening mechanisms of AlSi10Mg samples change before and after the various heat treatments affecting the mechanical properties, as previously discussed. This behavior is described by all contributions of the following equation [65,66]:(4)$$\sigma_{y}=\sigma_{f}+\sigma_{\mathrm{ss}}+\sigma_{\mathrm{HP}}+\sigma_{\mathrm{Or}}+\sigma_{p}+\sigma_{\mathrm{pre}}$$
where $\sigma_{f}$ is the lattice friction stress, $\sigma_{\mathrm{ss}}$ is the solid solution strengthening, $\sigma_{\mathrm{HP}}$ is the strengthening related to the grain size (Hall-Petch relationship), $\sigma_{\mathrm{Or}}$ is the Orowan strengthening related to the Si particles, $\sigma_{p}$ is the dislocation hardening, $\sigma_{\mathrm{pre}}$ is the contribute of the dislocations and precipitates. As reported by Li et al. [67], the first parameter can be neglected due to its small value (5.5 MPa) as compared with the other contributions, while these change significantly after the different heat treatments are undertaken. In the as-built conditions, without the precipitation phenomena induced by the preheated build platform, the strengthening mechanisms are described by the following parameters $\sigma_{\mathrm{ss}}$, $\sigma_{\mathrm{HP}}$ and $\sigma_{p}$. Increasing the amount of Si particles and the Mg2Si-phases by the precipitation phenomena (preheated build platform and/or DA at 200 °C/4 h), the $\sigma_{\mathrm{Or}}$ increases due to the Orowan looping mechanisms [46]. Hadadzadeh et al. [65] concluded that the Si particles are not shared by the dislocation, so, therefore, these contribute to a reinforcing phase. In the same context, after the DA at 200 °C/4 h, the strength decreases due to the decrease of solid solution strengthening and because of the dislocation density.

Firstly, Yang et al. [68] analyzed the following equation:(5)$$\sigma_{\mathrm{ss}}=k_{\mathrm{Mg}}{\left(C_{\alpha }^{\mathrm{Mg}}\right)}^{m}+k_{\mathrm{Si}}{\left(C_{\alpha }^{\mathrm{Si}}\right)}^{m}$$
where $k_{\mathrm{Mg}}$ and $k_{\mathrm{Si}}$ are 17 and 11 MPa wt%^−1^, m is 1 and C is the chemical concentration of Mg and Si [68,69]. They suggest that, after the DA and/or stress relief, Si and Fe will still be in solid solution as opposed to Mg, which will be present in a lower concentration due to its high diffusion rate. Finally, the $\sigma_{\mathrm{ss}}$ contribution can be considered negligible after the T6 heat treatment [68,69]. As m is 1, this suggests that this contribution can be neglected after the T6 HT. Secondly, Baek et al. [70] showed a high dislocation density in AlSi10Mg as-built samples than in the same samples after the direct aging at 180 °C for 6h. Initially, the rapid cooling induced by SLM process creates a high dislocation density which is partially eliminated by the DA. The same authors suggest that the direct aging heat treatment induces a uniform distribution of dislocation within the Al-matrix due to the precipitation phenomena. Finally, in the T6 heat-treated samples, the same authors highlighted the localization of density only around the coarse Si particles, and it is possible to conclude that the $\sigma_{p}$ decreases. Moreover, after this heat treatment the decrease of strength is also induced by the increase of grains size as described by the following Hall-Patch equation:(6)$$∆\sigma_{\mathrm{HP}}=\frac{K}{\sqrt{d}}$$
where K is the material constant, which is approximatively 0.04 MPa m^1/2^ [71], and d is the average grains size. As reported by Baek et al. [70], the $\sigma_{\mathrm{HP}}$ value decreases from 89 MPa in as-built condition to 12 MPa after the T6 HT.

The dislocation hardening can be estimated through the following equation [65]:(7)$$\sigma_{p}=\beta \mathrm{MGb}\sqrt{\rho_{p}}$$
where β is a constant of the material, M is the Taylor factor, G is the shear modulus of Al matrix, b is Burgers vector of Al and $\rho_{p}$ is the density of dislocation.

Straink et al. [72], who analyzed a model for predicting strength in overaged Al-Zn-Mg-Cu casting alloys, suggested a detailed relationship to determine the contribution of dislocation and precipitates:(8)$$\sigma_{\mathrm{pre}}=C_{4}\frac{\mathrm{Gb}}{{\left(l_{D}l_{t}\right)}^{1/2}}\;\left[f^{1/2}+0.70{\left(\frac{l_{D}}{l_{t}}\right)}^{1/2}f+0.12\left(\frac{l_{D}}{l_{t}}\right)f^{3/2}\right]$$
where $C_{4}$ is a constant, $l_{D}$ is the diameter of the discs of the precipitate, $l_{t}$ its thickness, and f is the volume fraction.

Figure 7a,c,e show the strain hardening rate (θ) as a function of true strain (ε) and the true stress-strain curves of all samples characterized by minimum and maximum elongation at break in as-built condition (Figure 7a, samples 4 and 14), after DA (Figure 7c, samples 1 and 21) and T6 HT (Figure 7e, samples 2 and 10). In this scenario, it is useful to highlight the Considère’s criterion, which defines the onset of necking through the intersection between the strain hardening rate and the true stress-strain curves [73]. The strain hardening rate (θ) is defined as the derivative of σ with respect to ε:(9)θ=dσ/dε
where σ and ε are the true stress and true strain, respectively. Figure 7b,d show the fracture surfaces of SL-90 sample in as-built condition and after the DA at 200 °C/4 h; while Figure 7f the fracture surface of SL-90 samples after the T6 HT. According to Considère’s criterion, the failure occurs before the necking formation for SL-90 samples in as-built condition and after the direct aging. In fact, the θ values are around 1800 and 1600 MPa, respectively. Chen et al. [46] show the same results considering only the as-built samples highlighting the strict correlation between the strain hardening rate and the Orowan strengthening mechanism, as previously mentioned. Focusing on Figure 7e, sample 10, which shows the minimum elongation at break value (Figure 6d), achieves the failure just before the necking formation, at θ of 335 MPa. Sample 2, however, shows the intersection between the true stress-strain and the strain hardening rate curves at 320 MPa and 0.101 of true stress and true strain, respectively. Considering the fracture surfaces shown in Figure 7b,d and that which was previously discussed, it is possible to observe the absence of necking and a fracture behaviour less ductile than sample 2 after the T6 HT (Figure 7f). Finally, the T6 heat-treated samples characterized by a ductility value over 12% (Figure 6d) show a failure that occurs after the necking formation rather than before.

Strain-hardening exponents (n) were calculated according to the ASTM standard E464 [74], and all obtained results are also confirmed by [44]. In this context, it is possible to emphasize that the n value calculated for the samples (2) is exactly 0.101 as previously reported (Figure 7e).

The strain hardening is an important value to understand concerning the material strengthening; in fact, it gives information about the plastic behaviour after the yield strength point. As widely reported in the literature, the n strain hardening exponent always assumes values between 0 and 1. More specifically, n is equal to 0 when the material is a perfectly plastic solid which shows the same value between the yield strength and the UTS, while the material is a perfectly elastic material when n = 1.

Figure 8a shows the strain hardening trends in function of the Z-axis considering the as built and the heat-treated 90-samples and 50-samples. Generally, the as-built samples are characterized by an increase of n values with increasing distance from the preheated build platform. This behaviour can be explained through the possible increase of high-density defects or dislocations as previously reported by the argumentation of the residuals stress [75]. Sjögren et al. [76] study the effective correlation between the dislocation movements and the strain hardening exponent (n), which increases with the increase of the multiplication and entanglement of dislocations. Moreover, the 90-samples show higher strain hardening exponents than the 50-samples, which tends to homogenize only after the DA at 200 °C/4 h. Also considering the top and the bottom region, the n values tend to stabilize due to the possible loss of the effects induced by the preheated build platform as reported in [23]. In detail, the DA can be induced by an increment of the precipitation phenomena both in top and bottom regions, and it can be considered as a stress relief HT for the top regions. In fact, the n values decrease with the possible decreasing of the dislocation density, as also reported by [77]. Finally, the n values decrease in all samples which were T6 heat-treated according to a decrease in the variation between the yield strength and the UTS values for each analyzed sample [78].

All of these can be justified by the microstructural variation (Figure 2 and Figure 4) such as by the grain size, second-phase particles, dislocation density and the SDAS [76,79]. As reported by Z. Li et al. [67], there is a strict relationship between the work strain hardening and the density mobile dislocation, which depends on the full-cellular and non-cellular structure. Moreover, the work hardening can assume a high value due to the high density of pre-existing dislocation, i.e., defects generated by the SLM process. In the full-cellular case, there are three possible scenarios: (1) dislocation de-penning from the supersaturated α-Al matrix; (2) a network of cutting dislocations at the interface Al-Si; and (3) dislocation emission from the interface Al-Si that moves through the grain. The same authors emphasize that the combination between the first and the second scenarios determine the hardening mechanisms. On the other hand, the mobile dislocation easily traverses the grains if the microstructure presents the fragmented and coarsened Si eutectic (non-cellular structure) as obtained after the T6 HT (Figure 4). Figure 8b reports the strain hardening rate normalized with the true stress [75] of the samples (4), (1) and (2) as representative of the as-built, direct aged and T6 heat-treated conditions, respectively. Focusing on as-built and direct aged samples, it is worth repeating that the failures occur before the necking formation, and these can be considered as premature failures caused by the mechanisms previously discussed. On the other hand, the T6 heat-treated sample shows a lower strain hardening rate than the other samples due to its microstructure. Moreover, sample (2) intersects the black dashed line, i.e., the value 1, which represents the points where the Considère’s criterion is satisfied. In fact, these values represent the equality between the true stress and the strain hardening rate.

Figure 9a shows the fracture profile of the sample (1) after DA at 200 °C/4 h, where it is possible to observe the non-linear course of the crack, in other words, a tortuous crack propagation through the microstructure. As matter of fact, the fracture profile shows some zones where the crack propagates in a straight line and other zones where it propagates following other geometries due to the propagation along the HAZs and coarse zones, as shown in Figure 9b,c. These zones, as discussed in Section 3.1., are generated along the boundaries of molten pools and laser scan tracks. Therefore, knowing that the laser scan tracks’ distribution and orientation are closely related to the scan strategy, this process parameter can affect the fracture mechanisms of samples characterized by a full-cellular microstructure.

The same results are obtained by Paul et al. [44] through the analysis of AlSi10Mg SLMed samples manufactured using different scan strategies. Focusing on Figure 9c, the white dotted line highlights a semi-ellipsoidal shape of the fracture profile, and observing the presence of the heat-affected zone, it can assume the presence of a laser scan track before failure of the sample. On the other hand, the cracks also propagated within both the molten pool (Figure 9d) and the laser scan tracks (Figure 9e), but the heat-affected zones remain the preferential areas of propagation.

In the same context, Delahaye et al. [43] show that the void initiation becomes more favorable along with the interface between the Al-matrix and coarsened Si particles in HAZs. As a matter of fact, considering the following equation, which describes the work (W) necessary to create a crack, it is possible to describe these fracture mechanisms as:(10)$$W\;\propto \gamma_{\mathrm{Al}}+\gamma_{\mathrm{Si}}-\gamma_{\mathrm{AlSi}}$$
where $\gamma_{\mathrm{Al}}$ and $\gamma_{\mathrm{Si}}$ are the surface energy of the Al-matrix and of the Si precipitates, respectively, while $\gamma_{\mathrm{AlSi}}$ is the interface energy. So, an increase of $\gamma_{\mathrm{AlSi}}$ due to the increase of Si eutectic and due to the tendency of de-cohesion between the particles and the matrix, the work to initiate a void (W) decreases, for example, from the center of the molten pool to the HAZ. On the other hand, the presence of pores can be the last, but not the least, cause of crack initiation, as shown in Figure 9e, where it is also possible to observe the cracked HAZ. As widely reported in the literature, depending on the quantity of defects present in AlSi10Mg SLMed samples, a competition between the brittle and the ductile fracture can be observed; namely, a crack propagation both within the molten pool or laser scan tracks and within the HAZs. As reported and discussed by Delahaye et al. [43], in the absence of defects, the crack propagation tends to occur along the HAZ, i.e., in the weakest zones of the microstructure.

The fracture profile of the T6 heat-treated sample (2) is shown in Figure 10a, where the orange arrow indicates a piece of fracture surface incorporated in epoxy resin (black areas). Also for this sample, the voids nucleate at the Si particles through the de-cohesion with the Al-matrix (Si particles show in Figure 10b); consequently, the void coalescence can occur as highlighted by the dotted circumference in Figure 10b, and the crack can propagate, as observed by Aboulkhair et al. [29]. During the tensile test, the crack propagation occurs, interconnecting the voids previously mentioned. In other words, it advances among the Si particles (Figure 10c). At the same time, the cracked Al-matrix is also observed in Figure 10c. Focusing on the optical microscope, the micrograph in Figure 10d shows the white dotted ellipse highlighting the possible presence of a molten pool before the T6 HT (see Figure 4c). In this context, it is possible to observe that the molten pool and the laser scan tracks, which remain after the HT as discussed in Section 3.1., can influence the fracture mechanisms. In fact, at their boundaries there is a greater presence of Si particles than at their center, so there are more possibilities that all voids can be interconnected by a crack, as reported by Zhao et al. [80].

In conclusion, to obtain an effective comparison between all analyzed samples, it is possible to use the Quality Index (QI) as well as the AlSi10Mg SMLed alloy as reported by Tang et al. [81]. This numerical parameter was defined by Drouzy et al. [82] for expressing the mechanical “quality” of the aluminium alloys, and it can be calculated through the following equation:(11)QI=UTS+dlog(ε)
where UTS is the ultimate tensile strength and d is an empirical parameter which is determined, as suggested by Drouzy et al. [82], to make QI independent of the yield strength, and ε is the engineering elongation to fracture. In this context, it is possible to observe that the as-built 90-samples and 50-samples are characterized by higher QI values (between 480–580 MPa) than the direct aged at 200 °C/4 h (between 460–525 MPa) as shown in Figure 11.

The same as-built QI values were calculated focusing on the studies carried out by Paul et al. [44] and Girelli et al. [25]. Both the bottom 90- and 50-samples show higher QI values than the top samples due to the highest ultimate tensile strengths, as reported in Figure 6a,b. On the other hand, the top 50-samples are characterized by the same quality indexes obtained after the DA at 200 °C/h. Furthermore, it can be concluded that this direct aging homogenized the mechanical properties along the build direction, and it may be a good compromise focusing on the QI values (460–540 MPa).

Finally, the T6 HT show the largest decrease of the QI range (415–470 MPa), and a higher homogenization than the other considered samples due to the marked microstructural variation (Figure 4). Considering the QI values analyzed by G. Sigworth [83], who studied the quality in aluminium casting, it possible to observe that the T6 heat-treated AlSi10Mg SLMed samples are perfectly comparable to the high-quality SS aluminum casting alloy.

## 4. Conclusions

In the study, AlSi10Mg samples manufactured by single e double laser selective laser melting using a layer thickness of 90 μm and a hatch spacing of 70 μm were analyzed before and after the direct aging (200 °C/4 h) and the T6 heat treatment (505 °C/4 h + 175 °C/4 h). In addition, these SL and DL samples have been compared to other AlSi10Mg selective laser melted samples, in the same heat-treated conditions, manufactured with a layer thickness lower than 40 μm and a hatch spacing higher than 100 μm. The main conclusions are as follows:The microstructure does not show any variation before and after heat treatments (direct aging 200 °C/4 h, T6) between the 50 and 90-samples in the same conditions, but the decrease of hatch spacing induces higher effects than the increase in layer thickness. In fact, the 90-samples are characterized by a finer microstructure formed by a mixture of columnar and equiaxed grains than the 50-samples despite the similar energy density values (∆φ = 6 J/(mm^3^)).The microstructure shows a great variation only after the T6 HT, where the Si eutectic network is destroyed and coarsened Si eutectic particles are formed. However, the ellipsoidal shape of the molten pools remains visible in the XY plane, and these could still influence the fracture mechanisms as well as in the samples directly aged. In the T6 heat-treated samples the crack propagates among the Si coarsened particles. Within the full-cellular structure, the crack propagation occurs along with the heat-affected zones and coarse zones.In as-built condition, the HV500 microhardness profile decreases along the build direction (Z-axis) from 130 ± 2 HV500 to 114 ± 4 HV500 and from 130 ± 2 HV500 to 117 ± 3 HV500 for SL-90 and DL-90, respectively, due to the precipitation hardening phenomena induced in the bottom region by the preheated build platform. In the same region, the high-density variation (12.20%) between the CM and the external frame of DL-90 induces a decrease of about 15 HV500, thus nullifying the precipitation hardening effects.The direct aging tends to homogenize the mechanical properties between top and bottom top, while at the same time inducing a small increase of ductility (from 6.9 ± 1.0% to 8.3 ± 0.9%). Only after the T6 HT did some samples show the elongation at break of 12–15%; while if the UTS values decreased approximately 294 ± 4 MPa, the yield strength values would remain constant with those obtained after direct aging (≈230 MPa). For these reasons, the strain hardening exponents decrease by approximately 0.24 for as-built and direct aged samples to 0.10 for T6 heat-treated samples.The Considère’s criterion shows that the failure occurs after the necking formation only in the samples characterized by a ductility higher than 12%. Moreover, the strain hardening rate was about 1800 MPa and 1600 MPa at the fracture point for the as-built and the direct aged samples, respectively.

## Figures and Tables

**Figure 1 materials-14-04901-f001:**
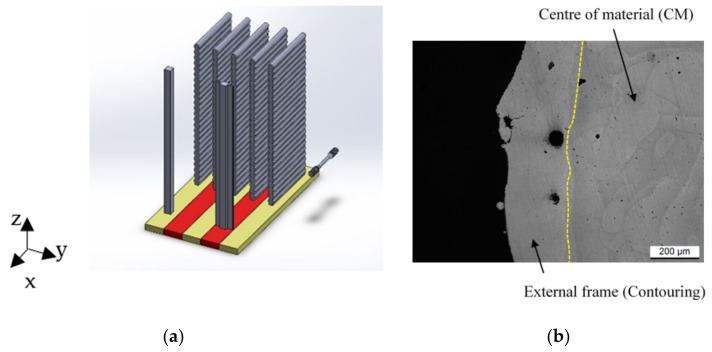
(**a**) Job of SL (yellow region) and DL (red region) AlSi10Mg 90-samples; (**b**) Optical microscope micrograph which shows the bar cross-section (⦿ Build direction).

**Figure 2 materials-14-04901-f002:**
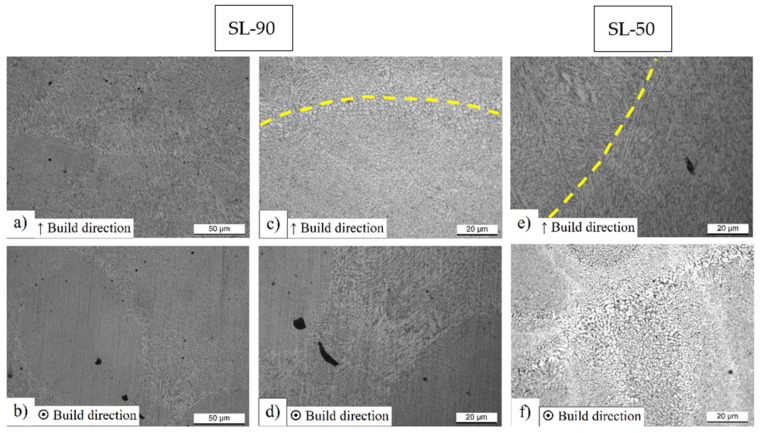
High magnification micrographs of as-built AlSi10Mg 90-samples in XZ plane (**a**,**c**) and XY plane (**b**,**d**), and of as-built AlSi10Mg 50-samples in XZ plane (**e**) and XY plane (**f**).

**Figure 3 materials-14-04901-f003:**
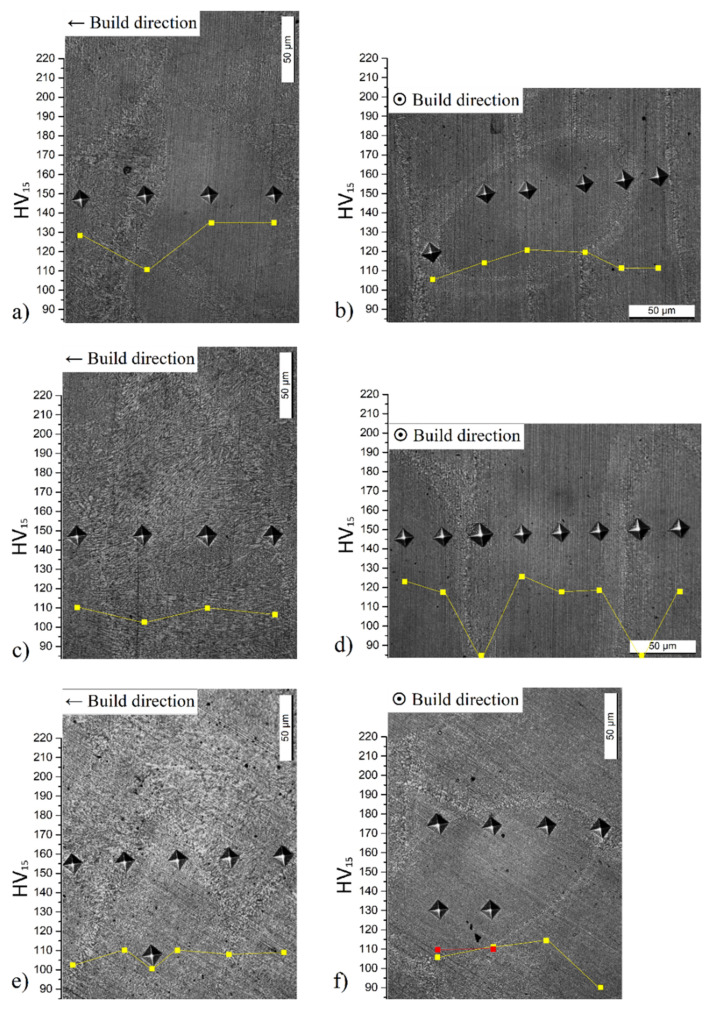
Vickers microhardness profiles (yellow and red curves) of as-built AlSi10Mg 90-samples (**a**–**d**) and 50-samples (**e**,**f**) performed in XZ (**a**,**c**,**e**) plane and XY plane (**b**,**d**,**f**).

**Figure 4 materials-14-04901-f004:**
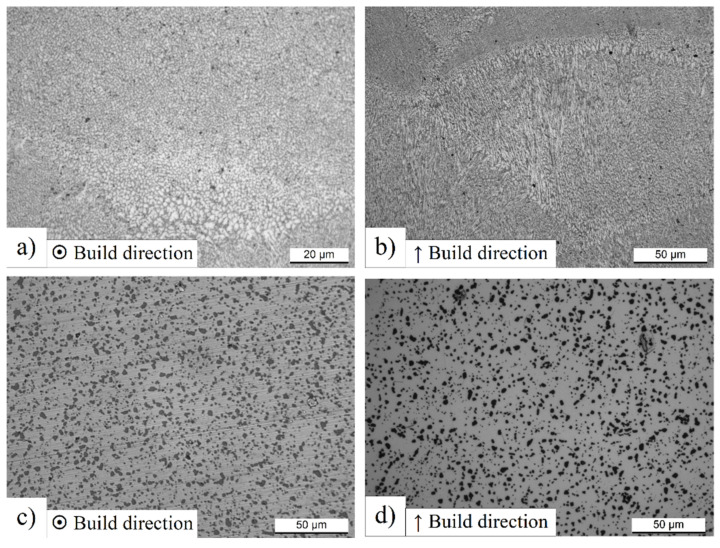
High magnification micrographs of SL AlSi10Mg DL-90 (**a**–**c**) and SL-50 (**d**) microstructure after 200 °C/4 h (**a**,**b**) and T6 (505 °C/4 h + 175 °C/4 h) HT (**c**,**d**). The first and second columns are referred to XY plane and XZ plane, respectively.

**Figure 5 materials-14-04901-f005:**
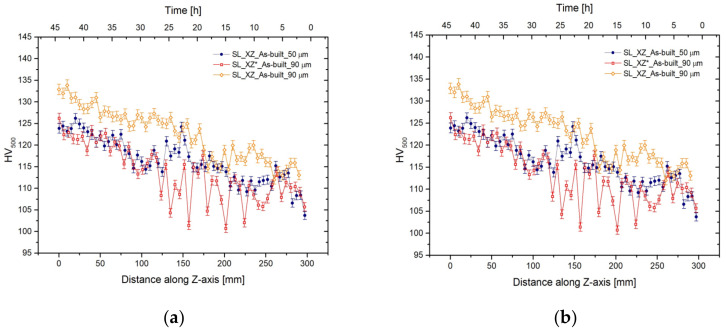
Microhardness profile of as-built SL (**a**) and DL (**b**) 90-samples carried out along the build direction (Z-axis). The blue and orange curves are related to the centre of 50-bars and 90-bars, respectively. The red curve represents the microhardness profile of the external frame of 90-samples.

**Figure 6 materials-14-04901-f006:**
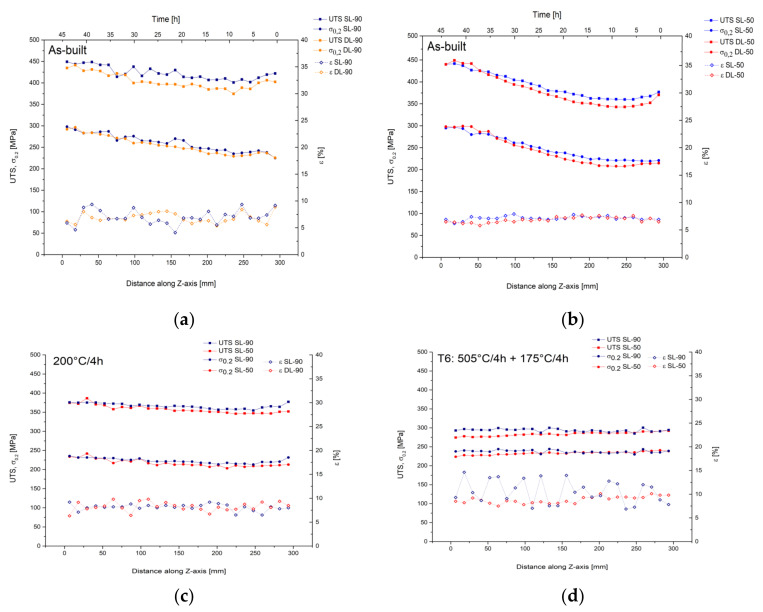
Tensile properties of AlSi10Mg samples: (**a**) as-built SL-90 and DL-90; (**b**) as-built SL-50 and DL-50; (**c**) SL-90 and SL-50 after 200 °C/4 h; (**d**) SL-90 and SL-50 after T6 heat treatment. The blue arrows indicate the samples will be considered in Figure 7.

**Figure 7 materials-14-04901-f007:**
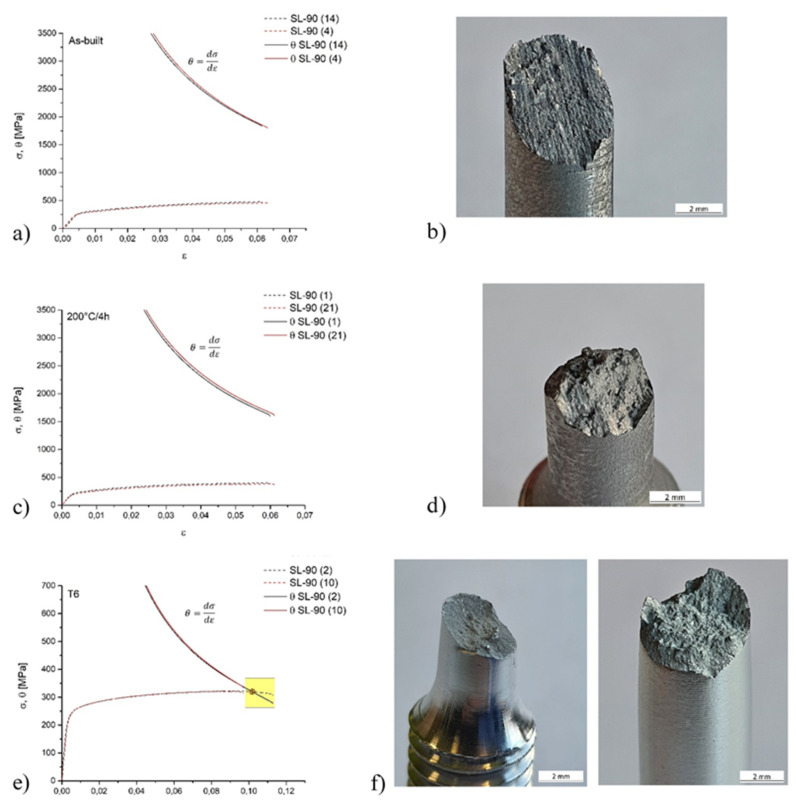
True tensile stress-strain curves and strain hardening rate θ (**a**,**c**,**e**) of SL-90 AlSi10Mg samples in as-built condition (**a**,**b**), after the direct aging (**c**,**d**) and the T6 heat treatment (**e**,**f**). Fracture surfaces of: (**b**) sample (4) in as-built condition, (**d**) sample (21) heat-treated at 200 °C/4 h, (**f**) samples (10) and (2) T6 heat-treated, respectively.

**Figure 8 materials-14-04901-f008:**
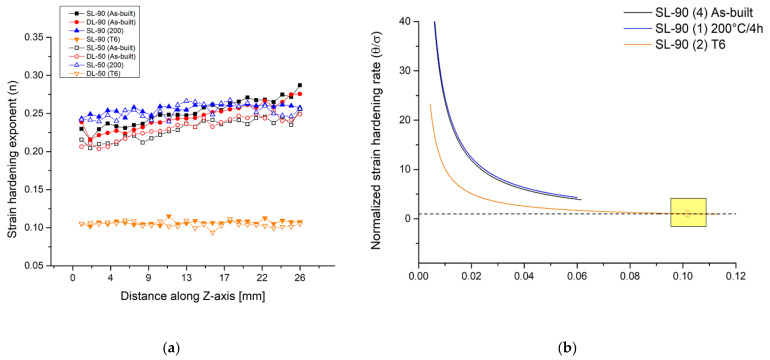
(**a**) Strain hardening values of as-built and heat-treated 90-samples and 50-samples in function of the distance from the build platform. (**b**) Normalized strain hardening rate θ=dσdεσ of as-built, direct aged and T6 heat-treated samples. The black dashed line indicates the points that satisfy the Considère’s criterion, in other words, where the strain hardening rate and the true stress coincide.

**Figure 9 materials-14-04901-f009:**
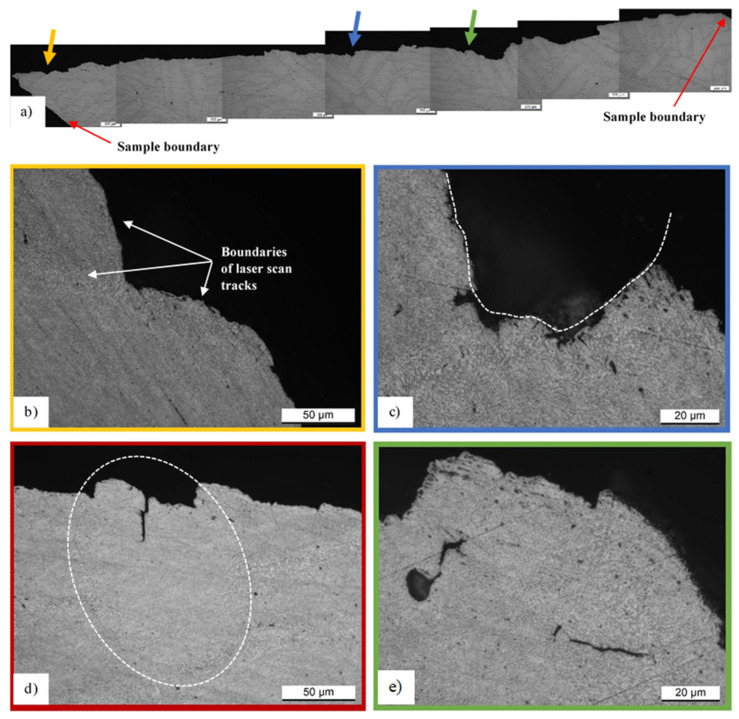
Fracture profile (**a**) of SL-90 (1) sample direct aged where the yellow, blue, red and green arrows indicate four different zones shown in (**b**–**e**) panels. The white dotted line (**c**) indicates a hypothetic laser scan track, while that in (**d**) indicates a molten pool.

**Figure 10 materials-14-04901-f010:**
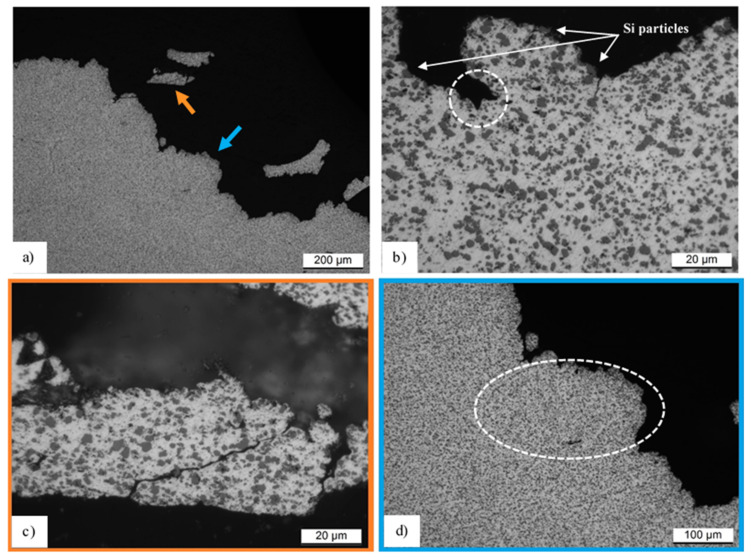
Fracture profile (**a**) of the T6 heat-treated sample (2) where the orange and light-blue arrows indicate two different zones shown in (**c**,**d**). The micrograph (**b**) shows another area of the same sample.

**Figure 11 materials-14-04901-f011:**
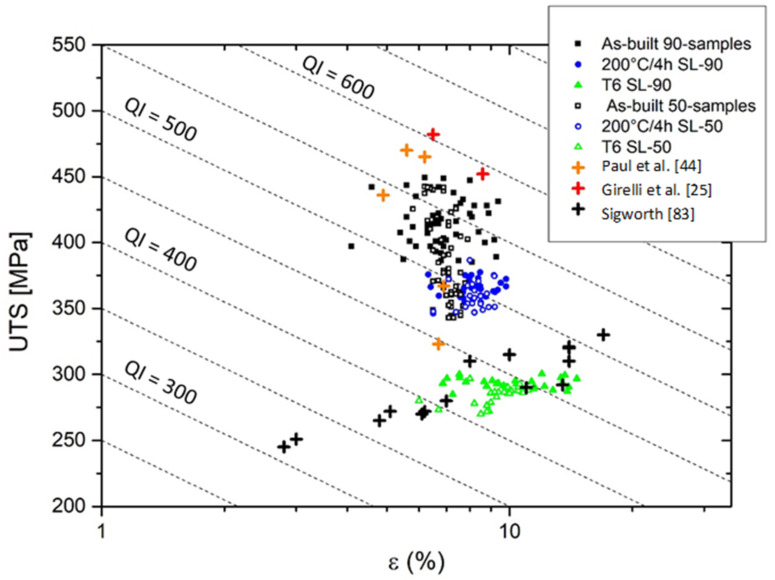
Comparison of mechanical properties of AlSi10Mg SLM in as-built and heat-treated conditions through the Quality Index (QI) with the results shown in the literature.

**Table 1 materials-14-04901-t001:** Chemical composition of AlSi10Mg samples (wt.%).

Elements	Al	Si	Fe	Mg	Cu	Mn	Zn	Ti	Pb	Sn
wt.%	Bal.	10.0	0.12	0.31	0.001	0.005	0.002	0.042	0.001	<0.01

**Table 2 materials-14-04901-t002:** Process parameters used for 50 and 90 AlSi10Mg samples.

Process Parameters	90-Samples	50-Samples
Layer thickness [μm]	90	50
Scan speed [mm/s]	1400	1150
Hatch spacing [μm]	70	170
Laser power [W]	370	350
Energy density [J/mm^3^]	42	36
Temperature of build platform [°C]	150	
Heat treatments	DA: 200 °C/4 h	
	T6: 505 °C/4 h + 175 °C/4 h	

**Table 3 materials-14-04901-t003:** Density values of AlSi10Mg 90-samples and 50-samples in as-built condition and after the heat treatments.

HT	Planes	SL		DL	
		Bottom	Top	Bottom	Top
As-built	XZ-90	99.62 ± 0.01%	99.76 ± 0.04%	99.20 ± 0.05%	99.33 ± 0.02%
	XZ-900 ^1^	97.74 ± 0.09%	97.72 ± 0.11%	86.99 ± 0.22%	95.50 ± 0.09%
	XZ-50	99.00 ± 0.01%	98.5 ± 0.03%	99.9 ± 0.01	99.2 ± 0.05%
200 °C/4 h	XZ-90	99.73 ± 0.02%	99.60 ± 0.01%	-	-
T6	XZ-90	98.15 ± 0.06%	97.95 ± 0.01%	97.58 ± 0.01%	98.12 ± 0.01%

^1^ These density values are referred to the external frame.

**Table 4 materials-14-04901-t004:** Variation of the mechanical properties from bottom to top of SL-90 and 50, DL-90 and 50 AlSi10Mg samples in as-built and heat-treated conditions.

HT		Absolute Differences			Relative Change		
		Δσ_0.2_	ΔUTS	Δσ_0.2_	ΔUTS	Δσ_0.2_	ΔUTS
As-built	SL-90	46 MPa	28 MPa	−0.4%	−16%	−6%	+6%
	DL-90	50 MPa	36 MPa	−0.6%	−18%	−8%	+8%
	SL-50 [24]	70 MPa	80 MPa	−0.8%	−24%	−18%	+12%
	DL-50 [24]	110 MPa	100 MPa	−0.9%	−37%	−22%	+14%
200 °C/4 h	SL-90	11 MPa	10 MPa	−0.5%	−5%	−3%	+6%
	SL-50	19 MPa	22 MPa	+0.8%	−8%	−6%	−10%
T6	SL-90	3 MPa	3 MPa	+1.8%	−1%	−1%	+16%
	DL-90	−11 MPa	−15 MPa	−1.6%	+5%	+5%	+20%

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
