# Peer review of "Work Hardening of Heat-Treated AlSi10Mg Alloy Manufactured by Selective Laser Melting: Effects of Layer Thickness and Hatch Spacing"

_materials, 2021, doi:10.3390/ma14174901_

Round 1
Reviewer 1 Report
Submitted manuscript entitled „Work hardening in heat-treated AlSi10Mg alloy manufactured by single and double laser selective laser melting: effects of layer thickness and hatch spacing“ brings knowledge from additive manufacturing area. The AlSi10Mg samples were manufactured by the SLM method with various building conditions and heat treatment of samples was carried out. The research was focused on microstructure, porosity and the mechanical properties were evaluated by hardness measurement and tensile test. The articles dealing with various building conditions and effects of subsequent heat treatment are necessary for using additive manufacturing in a wide application area.
Some revisions and explanations are needed.
Materials and methods: Size distribution of powder is really between 20 and 60 mm?
Figure 3: Are you sure that the distance between individual measuring points is sufficient to exclude deformation strengthening?
Page 14: I do not like the sentence „On the other hand, the variation between the CM (orange profile) and the external frame (red profile) is caused by the difference in terms of density as reported in Table 3.“ Hardness should be measured with suitable loading and positions. Pores should not affect hardness.
Which method was used for density (porosity) determination? The description of this method must be added in the section of Materials and methods.
What does mean time in X-axis in Figure 6a,b?
Reviewer 2 Report
The additive manufacturing of metallic parts is one of the fastest developing technology. The investigation of the microstructure and mechanical properties of the additively manufactured samples made from aluminium alloys is an actual topic. It may be especially interesting for the aluminium-silicon alloys, where the very fine eutectic microstructure is formed due to elevated cooling rates [10.1016/j.matlet.2018.11.033, 10.1007/s00170-015-7137-8]. In the paper "Work hardening in heat-treated AlSi10Mg alloy manufactured by single and double laser selective laser melting: effects of layer thickness and hatch spacing" the effect of the SLM parameters and position of the samples during printing on the microstructure and mechanical properties of the AlSi10Mg alloy was investigated. Using different technics, the authors carefully investigated the difference in the microstructure, hardness, and tensile properties of the printed samples. The authors have made well experimental work and discuss the results at a high scientific level. However, in my opinion, some points of the paper should be improved accordingly following comments:
- The additional information about specific values of the SLM (Table 2) and heat treatment parameters (DA and T6 HT temperature and time) should be added to the Materials and methods part. Why did the authors use 4 hours for annealing before quenching? A smaller time is enough for the dissolution of the excessive phases due to its small size.
- How was the experimental error of the microhardness measurement (±4) determined? As seen in Figure 3 only one imprint was made for each position.
- How was calculated the cooling rate for different SLM parameters? It is hard that such a small difference in the colling rate may have a significant influence on the microhardness. The applying of the Eqs. (2) and (6) for two presented cooling rates gives the difference in 7 MPa in terms of the stress (it correspondents only about 3 HV units).
- The additional information about density measurement should be added to the Materials and methods part.
- The contribution of the precipitates should be added to Eq. (3) by a separate term. The contribution of the dislocations and precipitates usually is not a simple sum but represents as root from the sum of its squares (please, see 10.1016/S1359-6454(03)00363-X).
- How was determined the true stress and true strain values for Figure 7? The calculation of the strain hardening rate seems to be incorrect. Its value for the maximum on the true stress – true strain curve should be zero. However, it has a large positive value (for example, about 300 MPa in Figure 7e).
- Minor changes are also required:
- It is better to use a template for the Metals articles;
- The paper should be carefully re-read to correct some typos and errors (e.g. page 19: σp is not dislocation density; page 20: instead of “whit” should be “with”, etc.)
Reviewer 3 Report
The paper entitled "Work hardening in heat-treated AlSi10Mg alloy manufactured by single and double laser selective laser melting: effects of layer thickness and hatch spacing" was presented for review. This work is devoted to the investigation of SLM process parameters on tensile properties and microstructural variations.
Small issues:
1. The MDPI journal template should be used for submission.
2. I suppose that abstract is too long. Please shorten it.
3. Fig. 1b - please specify build direction in the image.
4. Please specify the equipment used for Vickers microhardness measurements.
5. p.12 "SSS (solid supersaturated solution)" no need in the double description of the abbreviatures.
In general, it is well-written research with a deep and comprehensive discussion of the state of the art in SLM of aluminum alloys, applied heat treatments, and microstructural mechanisms.
Round 2
Reviewer 2 Report
The authors have partially answered previous comments. The paper is needed to be modified accordingly following comments:
- The authors have described why they used a time of 4 hours for DA. However, the question was: why did the authors use 4 hours for annealing before quenching at 505 °C? A smaller time is enough for the dissolution of the excessive phases due to its small size. A smaller time of annealing may give a higher level of mechanical properties.
- The authors answered, that the standard deviations were obtained by all microhardness values that form each profile. It seems to be incorrect. The standard deviation should be determined using several measurements at each position in the profile. However, as seen in Figure 3 only one imprint was made for each position.
- The authors have inserted the equation for precipitation hardening. However, they did not modify Eq. (3) (in the current version Eq. (4). The contribution of the precipitates should be added to Eq. (3) by a separate term. The contribution of the dislocations and precipitates usually is not a simple sum but represents as root from the sum of its squares (please, see 10.1016/S1359-6454(03)00363-X).
- Figure 7 seems to be incorrect. The true stress cannot decrease during the deformation and the value of the strain hardening rate for the maximum on the true stress – true strain curve should be zero. The authors have answered that the derivative is zero only if the stress and the strain are constant. It is strongly incorrect. The derivative has zero value if the true stress is constant while true strain changes. The authors should carefully correct their calculations.
Round 3
Reviewer 2 Report
The manuscript may be accepted for publication.
Author Response
thank you for your revision. It seems there is nothing to add.
Regards,
E. Cerri